# Is Composition of Brain Clot Retrieved by Mechanical Thrombectomy Associated with Stroke Aetiology and Clinical Outcomes in Acute Ischemic Stroke?—A Systematic Review and Meta-Analysis

Joanna Huang [1,2,3], Murray C. Killingsworth [3,4,5] and Sonu M. M. Bhaskar [1,2,5,6,7,8,*]

1   Global Health Neurology Lab, Sydney, NSW 2000, Australia
2   Neurovascular Imaging Laboratory, Ingham Institute for Applied Medical Research, Clinical Sciences Stream, Sydney, NSW 2170, Australia
3   UNSW Medicine and Health, University of New South Wales (UNSW), South Western Sydney Clinical Campuses, Sydney, NSW 2170, Australia
4   Department of Anatomical Pathology, NSW Health Pathology, Correlative Microscopy Facility, Ingham Institute for Applied Medical Research and Western Sydney University (WSU), Liverpool, NSW 2170, Australia
5   NSW Brain Clot Bank, NSW Health Pathology, Sydney, NSW 2170, Australia
6   Liverpool Hospital & South West Sydney Local Health District (SWSLHD), Department of Neurology & Neurophysiology, Sydney, NSW 2170, Australia
7   Ingham Institute for Applied Medical Research, Stroke & Neurology Research Group, Sydney, NSW 2170, Australia
8   Clinical Sciences Building, 1 Elizabeth St., Liverpool Hospital, Liverpool, NSW 2170, Australia
*   Correspondence: sonu.bhaskar@globalhealthneurolab.org; Tel.:+61-(02)-8738-9179; Fax: +61-(02)-8738-3648

**Abstract: Background:** Brain clots retrieved following endovascular thrombectomy in acute ischemic stroke patients may offer unique opportunities to characterise stroke aetiology and aid stroke decision-making in select groups of patients. However, the evidence around the putative association of clot morphology with stroke aetiology is limited and remains inconclusive. This study aims to perform a systematic review and meta-analysis to delineate the association of brain clot composition with stroke aetiology and post-reperfusion outcomes in patients receiving endovascular thrombectomy. **Methods:** The authors conducted a systematic literature review and meta-analysis by extracting data from several research databases (MEDLINE/PubMed, Cochrane, and Google Scholar) published since 2010. We used appropriate key search terms to identify clinical studies concerning stroke thrombus composition, aetiology, and clinical outcomes, in accordance with Preferred Reporting Items for Systematic Reviews and Meta-Analyses (PRISMA) guidelines. **Results:** The authors identified 30 articles reporting on the relationship between stroke thrombus composition or morphology and aetiology, imaging, or clinical outcomes, of which 21 were included in the meta-analysis. The study found that strokes of cardioembolic origin (SMD = 0.388; 95% CI, 0.032–0.745) and cryptogenic origin (SMD = 0.468; 95% CI, 0.172–0.765) had significantly higher fibrin content than strokes of non-cardioembolic origin. Large artery atherosclerosis strokes had significantly lower fibrin content than cardioembolic (SMD = 0.552; 95% CI, 0.099–1.004) or cryptogenic (SMD = 0.455; 95% CI, 0.137–0.774) strokes. Greater red blood cell content was also significantly associated with a thrombolysis in cerebral infarction score of 2b–3 (SMD = 0.450; 95% CI, 0.177–0.722), and a positive hyperdense middle cerebral artery sign (SMD = 0.827; 95% CI, 0.472–1.183). No significant associations were found between red blood cell, platelet, or white blood cell content and aetiology, or between clot composition and bridging thrombolysis. **Conclusions:** This meta-analysis found that fibrin composition is significantly higher in strokes of cardioembolic and cryptogenic origin, and that red blood cell content is positively associated with the hyperdense middle cerebral artery sign and better reperfusion outcomes. Important advances to stroke clinical workup can be derived from these findings, in which many aspects of stroke workflow remain to be optimised. As data are still limited in terms of the association of various thrombus components with stroke aetiology as well as a standardised method of analysis, further studies are required to validate these findings to guide their use in clinical decision-making.

**Keywords:** stroke; reperfusion therapy; thrombectomy; etiology; clot morphology; clot composition; brain clot

## 1. Introduction

Stroke is the second leading cause of death and third leading cause of disability worldwide [1], with acute ischaemic stroke (AIS) accounting for approximately 80% of this burden [2,3]. With less than 15% of patients able to receive intravenous thrombolysis (IVT) [4–8], the advent of endovascular thrombectomy (EVT) has dramatically improved stroke outcomes in eligible patients, with the publication of five randomised control trials in 2015 [9–13] leading EVT to be incorporated into the standards of care. EVT has also provided the opportunity to examine retrieved clots, offering an avenue to determine associations between clot morphology and the aetiology of underlying stroke [14].

Clot morphology and its histopathological characterisation has gained tremendous research and clinical interest in stroke medicine, in the last 7 years or so, since 2015 [14,15]. Despite emerging evidence, the links between clot histopathology and stroke aetiology and outcomes remain unclear. For example, while several studies indicate a link between red blood cell (RBC) content and large artery atherosclerosis (LAA) strokes [16–21], some reported no statistically significant findings [22–24], whilst others found contrasting findings [25,26]. Notably, cardioembolic stroke has been traditionally known to be rich in RBCs, as reported in earlier studies [25]; however, recent studies with large sample size have contested these findings [14,17,18]. Evidently, there is outstanding ambiguity around the putative association of specific stroke aetiology and clot composition. This is especially poignant to cryptogenic strokes, which contributes to 30–40% of AIS patients [27], and poses a huge diagnostic challenge in a real-world setting. Interestingly, a recent meta-analysis from our group also reported significant association of LAA or cardioembolic stroke aetiologies with collateral status in AIS patients receiving reperfusion therapy, with LAA and cardioembolic being associated with increased rate of good and bad collaterals, respectively [28]. Beyond embolism, other factors such as atrial fibrillation and human immunodeficiency virus (HIV) infection may also cause stroke [29,30]. Improving understanding of aetiology, especially strokes of undetermined cause or cryptogenic stroke, as well as its association with pre-intervention imaging signs such as Hyperdense Middle Cerebral Artery Sign (HMCAS), could help optimise management workflows regarding procedural options and ongoing treatment.

It is imperative to critically investigate evidence around the link between clot morphology and stroke aetiology or outcome. This study aims to investigate the association between brain clot composition with the aetiology of stroke and outcomes after EVT by performing a meta-analysis of individual studies.

Our underlying questions are, in AIS patients receiving EVT:

(1)    Is clot composition associated with stroke aetiology?
(2)    Is clot composition associated with successful recanalisation?
(3)    Is clot composition associated with the pre-interventional HMCAS? and
(4)    Does bridging thrombolysis influence brain clot composition following EVT?

## 2. Methods

### 2.1. Literature Search: Identification and Selection of Studies

This study was performed following the Preferred Reporting Items for Systematic Reviews and Meta-Analyses (PRISMA) guidelines. Published studies (from 1 January 2010 to 1 August 2022) were retrieved primarily using the databases MEDLINE/PubMed. Additionally, Google Scholar database was also searched using the combination of keywords. Cochrane Library and said databases were searched for systematic reviews, meta-analysis and relevant additional references were retrieved for inclusion in this analysis. Keywords used in the search included stroke, thrombus, thrombectomy, clot retrieval and clot compo-

sition. In addition, relevant references were reviewed to retrieve extra studies for inclusion in this analysis. The full search strategy is provided in the Supplementary Information.

### 2.2. Inclusion and Exclusion Criteria

Studies were eligible if they met the following criteria: (1) Patients aged 18 years or above; (2) Patients diagnosed with AIS; (3) Patients who received EVT; (4) Studies with good methodological design; and (5) Studies where clot characteristics and prognosis after endovascular thrombectomy are available. The exclusion criteria were: (1) Animal Studies; (2) Duplicated Publications; (3) Full-Text Article not available; (4) Systematic Reviews, Meta-Analyses, Conferences, Letters, and Case Reports or Series; (5) Studies with a histological cohort under 20 patients, and (6) Studies with relevant data on clot histology unavailable or no related outcome measured.

### 2.3. Data Extraction

Firstly, titles and abstracts were screened on EndNote 20 (Clarivate, PA, USA), and studies were excluded according to the eligibility criteria specified above. The main texts of the remaining articles were thoroughly reviewed, and studies were either included or excluded depending on the eligibility criteria, provided data were available. Published reviews, past meta-analyses, opinions, and other relevant articles were reserved for discussion. Two authors independently screened the articles and discussed and came to an agreement on any discrepancies. The data from each study were extracted independently using a data extraction sheet to obtain the following information on: (1) baseline demographics, author and year of publication; (2) study population: sample size, patient characteristics, clinical variables (stroke aetiology, thrombolysis in cerebral infarction (TICI) score, HMCAS, bridging therapy); (3) outcome variables: the clot composition (RBC, fibrin, platelet, white blood cell (WBC) fraction) of extracted thrombi as determined on histological analysis, and (4) adverse effects. Data were extracted from graphs with precision, where raw data were not available.

### 2.4. Quality Assessment of Included Studies

The methodological quality of each study was assessed independently by two researchers using the Modified Jadad Scale (Supplementary Information SI Table S1). The scale assesses the quality of studies according to 8 criteria evaluation criteria. The risk of publication bias was also assessed using the following scoring system: A score of 0 indicates a low potential for bias, a score of 1–2 indicates a moderate potential for bias (1: any conflicts of interest declared relating to industry funding outside of the current research publication; 2: if the study had industry funding) and score 3 indicates a high potential for bias. The absence of industry funding did not necessarily convey an absence of bias.

### 2.5. Statistical Analysis

Statistical analyses were performed using Stata (Version 13.0, StataCorp, College Station, TX, USA). A random effects meta-analysis on clot composition and aetiology, and various clinical variables, was performed using the "metan" package, generating a Standard Mean Difference (SMD) and forest plot for each hypothesis. For studies which provided data in median (IQR) form, Wan's Method was used to convert these data into mean (SD) form [31]. Egger's test of effect sizes was performed in each meta-analysis to investigate potential publication bias. Finally, the influence of a single study in meta-analysis estimation was examined using the "metaninf" STATA package.

### 2.6. Investigations of Heterogeneity

Heterogeneity was studied in each meta-analysis, assessed using Cochran's Q test for heterogeneity and expressed as the $I^2$ index (0–49% = low, 50–74% = moderate, >75% = high).

## 3. Results

### 3.1. Results of the Search

The initial search yielded 315 entries. After 20 duplicates were removed, and 34 studies were added from additional references/sources, 329 studies were left for initial screening. Based on titles and abstracts, 267 studies were excluded and 62 remained. Reviewing these studies' full texts excluded another 32 papers (as displayed in the PRISMA flowchart in Figure 1), resulting in 30 studies that satisfied the inclusion criteria for systematic review. A further nine studies were excluded (Supplementary Information SI Table S2), resulting in 21 studies, comprising of 2468 patients, for the final quantitative meta-analysis (Table 1).

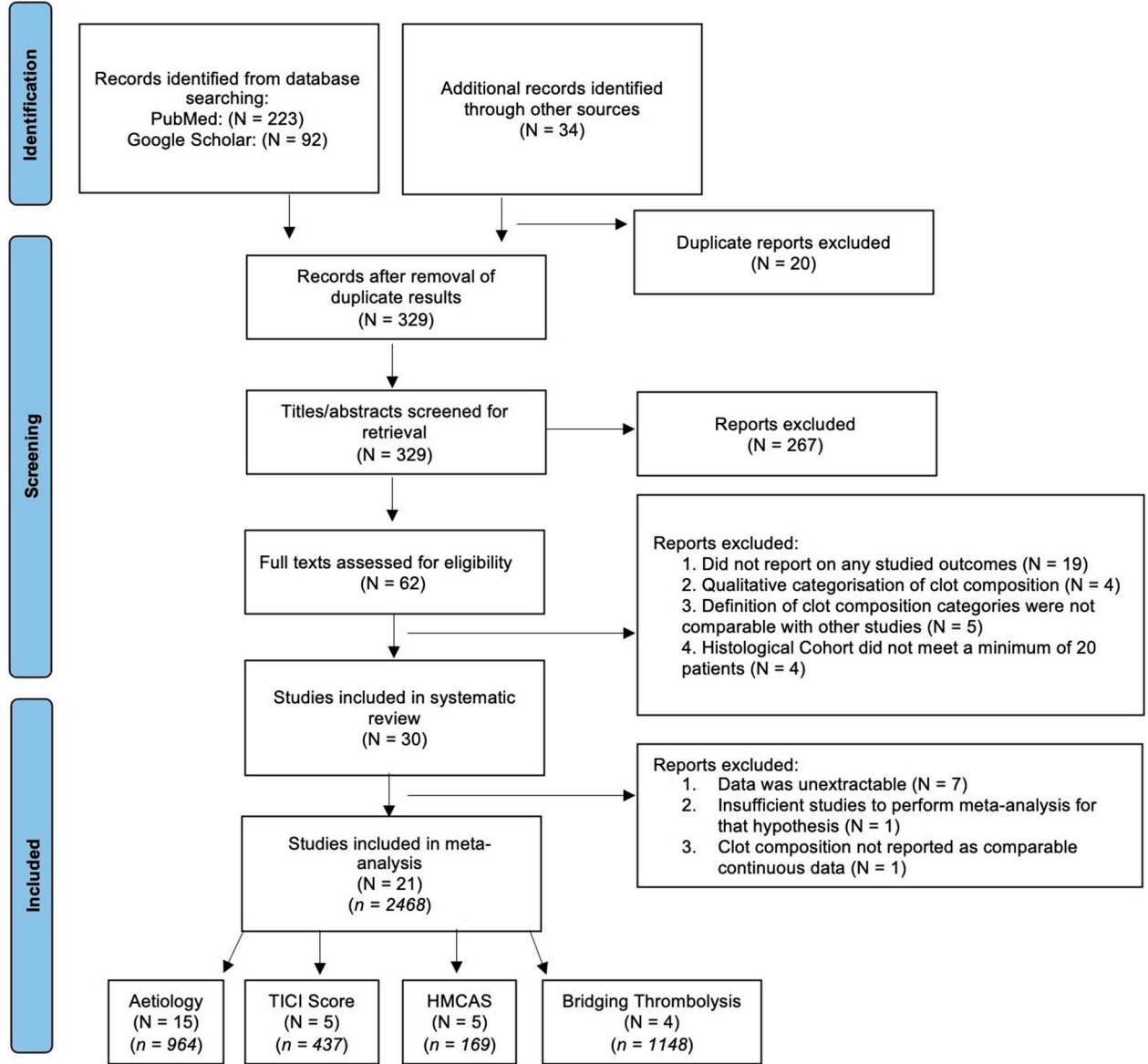

**Figure 1.** PRISMA Flowchart. Abbreviations: TICI: Thrombolysis in Cerebral Infarction scale, HMCAS: Hyperdense Middle Cerebral Artery Sign, N: Number of Studies, *n*: Number of Patients.

### 3.2. Study Characteristics

Of the included studies for meta-analysis, 15 studies [16–26,32–35] reported on the relationship between brain clot composition and stroke aetiology (Table 2). All studies used the Trial of Org 10,172 in Acute Stroke Treatment (TOAST) classification system, which includes large artery atherosclerosis (TOAST 1), cardioembolism (TOAST 2), small artery occlusion (TOAST 3), strokes of other determined aetiology (TOAST 4) and strokes

of undetermined aetiology (TOAST 5), except one study [32] which did not specify its classification method. Non-cardioembolic strokes were dichotomised as TOAST 1 + TOAST 4 strokes, except in three studies [24–26] which reported zero TOAST 4 strokes. Since there was no mention of exclusion, it was taken that no TOAST 4 strokes occurred and thus non-cardioembolic was defined as TOAST 1 only for these studies. Data were collected for comparison between cardioembolic and non-cardioembolic strokes, as well as between cardioembolic and LAA strokes. Successful recanalisation was defined as achieving TICI score of 2b–3. Data were collected on the clot component fractions in these two groups from five studies [16,26,36–38] (Table 3). Pre-interventional imaging signs comprised of the HMCAS as seen on CT, and susceptibility vessel sign (SVS) as seen on magnetic resonance imaging. Data were collected from five studies on the relationship between the HMCAS and clot composition (Table 4). Lastly, bridging thrombolysis was defined as the administration of IVT using recombinant tissue plasminogen activator (r-tPA) prior to EVT. Data were collected from four studies on the association between clot composition and bridging thrombolysis [13,19,21,34] (Table 5).

Hematoxylin and eosin (H&E) staining was the most commonly performed histological analysis, used by 19 studies [16–23,25,26,32–40]. Martius Scarlet Blue (MSB) [16,22–24,40] and Elastica van Gieson (EVG) [18,28,30,32,33,35] staining were both selected in 5 studies, Prussian Blue in 3 [18,28,33,35], Masson's trichrome in 2 [36,37], and Ladewig's trichrome [39], Von Kossa [39], naphthol AS-D [39], chloroacetate [39], and Mallory's phosphotungstic acid-hematoxylin [18] were included once each. One study used scanning electron microscope [32], and eight studies employed immunohistochemistry in their analyses [13,15,16,18,22,29,33,35,36].

### 3.3. Association between RBC Content and Aetiology

A total of 14 studies reported on the association between RBC content and stroke aetiology [16–26,33,41,42]. RBC content was found to be higher in cardioembolic strokes in three studies [22,23,37], non-cardioembolic or LAA strokes in five studies [16–21], and no statistical difference in five studies [22–24,33,42]. Nine studies containing 537 participants had data available for meta-analysis comparing RBC content in strokes of non-cardioembolic and cardioembolic origin [17–21,23–26]. This revealed that RBC content was greater in strokes of non-cardioembolic aetiology relative to cardioembolic aetiology (SMD = 0.184; 95% CI, $-0.191$–0.558, $p = 0.337$), though this result was not statistically significant. There was moderate heterogeneity ($I^2 = 71.5\%$, $p < 0.001$), and Egger's test revealed significant publication bias (Supplementary Information SI Table S3). A second meta-analysis, including 10 studies [16–20,22–26] containing 446 subjects compared RBC content in strokes of LAA and cardioembolic origin, concluded that the greater RBC content was associated with LAA strokes in comparison to cardioembolic strokes (SMD = 0.368; 95% CI, $-0.138$–0.874, $p = 0.154$); however, this association was not statistically significant. Heterogeneity was high ($I^2 = 80.0\%$, $p < 0.001$), and a random-effects model was used. Egger's test found no significant publication bias (Supplementary Information SI Table S3).

### 3.4. Association between Fibrin and Aetiology

A total of 13 studies reported on the association between fibrin content and stroke aetiology [16–19,21–25,32–34,41]. Fibrin content was greater in cardioembolic strokes in five studies [13,14,16,18,27], non-cardioembolic or LAA strokes in two studies [25,41], and not statistically significant in six studies [18,22–24,33,34]. For the meta-analysis comparing fibrin content in cardioembolic and non-cardioembolic strokes, seven studies with 395 patients were included [17–19,21,23,25,34]. The meta-analysis demonstrated a significantly greater percentage of fibrin in patients with cardioembolic strokes, in comparison with non-cardioembolic strokes (SMD = 0.388; 95% CI, 0.032–0.745, $p = 0.033$). The heterogeneity was moderate ($I^2 = 56.0\%$, $p = 0.034$), and Egger's test did not find any publication bias (Supplementary Information SI Table S3). Eight studies [16–19,22,23,25,32,34] with 328 patients were included in the meta-analysis comparing fibrin content in strokes of cardioembolic

and LAA origin, also showing significant association between fibrin content and cardioembolic strokes (SMD = 0.552; 95% CI, 0.099–1.004, *p* = 0.017). Heterogeneity was moderate ($I^2$ = 67.9%, *p* = 0.003), and a random-effects model was used. Egger's test demonstrated significant publication bias in this meta-analysis (Supplementary Information SI Table S3).

### 3.5. Association between Platelet Content and Aetiology

Seven studies [16,18,19,23–25,39] examined the association between platelet content and stroke aetiology. One study [23] found a greater platelet proportion in strokes of LAA origin compared to cardioembolic strokes, while the other six studies [13,15,16,21,22,35] found no significant association. Six studies [13,15,16,20,22,35] with 284 participants were included in the meta-analysis studying platelet content and aetiology. There was no significant difference in platelet fraction found between cardioembolic and LAA strokes (SMD = 0.168; 95% CI, −0.360–0.696, *p* = 0.533). There was moderate heterogeneity ($I^2$ = 72.9%, *p* = 0.002), and a random-effects model was used. Egger's test found significant publication bias (Supplementary Information SI Table S3).

### 3.6. Association between WBC Content and Aetiology

A total of 10 studies investigated the association between WBC content and stroke aetiology [16,17,20–23,25,26,33,35]. One study [26] reported greater WBC content in strokes of non-cardioembolic origin, and another study [21] reported greater WBC content in cardioembolic strokes. Seven studies found no significant association [13,14,17,19,20,22,28,30]. Eight studies with 339 participants were included in the meta-analysis studying WBC content and aetiology. There was no significant difference in WBC proportion found between cardioembolic and LAA strokes (SMD = −0.028; 95% CI, −0.394–0.338, *p* = 0.110). The heterogeneity was low ($I^2$ = 48.2%, *p* = 0.061), and Egger's test demonstrated significant publication bias (Supplementary Information SI Table S3).

### 3.7. Association between RBC Content and Cryptogenic Stroke

Twelve studies reported on the association between RBC content and strokes of cryptogenic origin [16–20,22,24–26,33,38,39]. Three studies found an association between non-cardioembolic or LAA strokes and greater RBC content [13,15,33], and nine studies found no significant difference [17,19,20,22,24–26,33,39]. Seven studies with 272 participants were included in the meta-analysis comparing RBC content between cryptogenic and non-cardioembolic stroke groups. Cryptogenic strokes had lower RBC fraction, when compared with non-cardioembolic strokes (SMD = −0.232; 95% CI, −0.651–0.188, *p* = 0.280), though this result was not statistically significant. There was moderate heterogeneity ($I^2$ = 55.2%, *p* = 0.037), and Egger's test revealed significant publication bias (Supplementary Information SI Table S3). There were 10 studies with 262 participants included in the meta-analysis comparing RBC content in cryptogenic and LAA stroke patients. The meta-analysis demonstrated lower RBC content in cryptogenic strokes relative to LAA strokes (SMD = −0.336; 95% CI, −0.738–0.065, *p* = 0.100); however, this result was not statistically significant. There was moderate heterogeneity ($I^2$ = 54.7%, *p* = 0.019), and random-effects modelling was used. Egger's test did not find significant publication bias (Supplementary Information SI Table S3).

### 3.8. Association between Fibrin Content and Cryptogenic Stroke

A total of 10 studies reported on the association between fibrin content and cryptogenic stroke [16–19,22,23,25,34,38,39]. Four studies found an association between fibrin content and cardioembolic strokes [13,16,29,33], and six studies found no significant difference [14,15,19,20,22,35]. Six studies with 252 participants were included in the meta-analysis comparing fibrin content between cryptogenic and non-cardioembolic stroke groups [15,16,20,22,29,33]. Fibrin content was found to be significantly greater in cryptogenic strokes relative to strokes of non-cardioembolic origin (SMD = 0.468; 95% CI, 0.172–0.765, *p* = 0.002). Heterogeneity was low ($I^2$ = 13.4%, *p* = 0.329) and Egger's test

demonstrated significant publication bias (Supplementary Information SI Table S3). There were seven studies with 168 participants included in the meta-analysis comparing fibrin content in cryptogenic and LAA stroke patients [16–19,22,23,25]. The meta-analysis demonstrated a significantly higher fibrin percentage in cryptogenic strokes relative to LAA strokes (SMD = 0.455; 95% CI, 0.137–0.774, *p* = 0.005). There was low heterogeneity ($I^2$ = 0.5%, *p* = 0.420). Egger's test did not find significant publication bias (Supplementary Information SI Table S3).

### 3.9. Association between Platelet Content and Cryptogenic Stroke

A total of seven studies [16,18,19,23–25,39] reported on the association between platelet content and cryptogenic stroke. Two studies found greater platelet proportion in cryptogenic strokes compared to LAA strokes [16,18], while the other five studies found no significant difference [19,23–25,39]. There were six studies with 152 participants included in the meta-analysis comparing platelet content in strokes of cryptogenic and LAA origin [13,15,16,20,22,35]. The meta-analysis found no statistically significant association between platelet proportion and stroke origin (SMD = −0.001; 95% CI, −0.669–0.666, *p* = 0.997). There was moderate heterogeneity ($I^2$ = 71.7%, *p* = 0.003), and random-effects modelling was used. Egger's test demonstrated no significant publication bias (Supplementary Information SI Table S3).

### 3.10. Association between WBC Content and Cryptogenic Stroke

Ten studies reported on the association between WBC content and cryptogenic stroke [16,17,20–23,25,26,33,35]. One study found an association between WBC content and cryptogenic stroke [21], while the other nine studies found no significant association [13,14,17,19,20,22,23,28,30]. For the meta-analysis comparing WBC content between cryptogenic and LAA stroke groups, eight studies with 204 participants were included [13,14,17,19,20,22,23,30]. The meta-analysis demonstrated greater WBC content in cryptogenic strokes in comparison to LAA strokes (SMD = 0.227; 95% CI, −0.057–0.511, *p* = 0.117); however, this result was not statistically significant. There was low heterogeneity ($I^2$ < 0.1%, *p* = 0.502). Egger's test demonstrated significant publication bias (Supplementary Information SI Table S3).

### 3.11. Association between Clot Composition and Successful Recanalisation

A total of eight studies reported on the association between clot composition and successful recanalisation [16,26,36–38,43–45]. Single studies found an association between greater RBC content [37] or platelet content [44] and successful recanalisation, and one study found higher platelet content to be associated with unsuccessful recanalisation [43]. The other five studies reported no significant association [13,23,31,33,41]. Five studies with 437 participants were included in the meta-analysis investigating RBC content and successful recanalisation [16,26,36–38]. This meta-analysis demonstrated a significant positive association between RBC percentage and successful recanalisation (SMD = 0.450; 95% CI, 0.177–0.722, *p* = 0.001). The heterogeneity was low ($I^2$ < 0.1%, *p* = 0.878), and Egger's test demonstrated significant publication bias (Supplementary Information SI Table S3).

### 3.12. Association between Clot Composition and Pre-interventional Imaging Signs

A total of 10 studies reported on the association between clot composition and pre-interventional imaging signs [16,25,26,33,35,40,46–49]. A positive HMCAS was associated with greater RBC content in three studies [23,28,36], greater platelet content in one study [46], and no significant associations were found in two studies [16,35]. A positive SVS was associated with greater RBC content in two studies [25,48], and one study [49] found no significant association. There were five studies with 169 participants included in the meta-analysis investigating RBC content and the HMCAS [13,23,28,30,36]. This meta-analysis demonstrated a significantly greater RBC fraction in patients with a positive HMCAS (SMD = 0.827; 95% CI, 0.472–1.183, *p* < 0.001). The heterogeneity was low ($I^2$ < 0.1%, *p* = 0.865),

and Egger's test demonstrated significant publication bias (Supplementary Information SI Table S3).

### 3.13. Influence of Bridging Thrombolysis on RBC Content

A total of seven studies investigated the influence of bridging thrombolysis on RBC content [13,19,21,34,43,45,46]. Singular studies found greater RBC proportion [22] and lower RBC proportion [49] to be associated with bridging thrombolysis; however, five studies reported no significant association [13,21,34,43,46]. Four studies with 1148 participants were included in the meta-analysis comparing RBC content in patients who received bridging thrombolysis vs. direct EVT [13,19,21,34]. The meta-analysis found that the RBC proportion was greater in patients who received bridging thrombolysis (SMD = 0.138; 95% CI, −0.109–0.385, $p = 0.274$); however, this result was not statistically significant. Heterogeneity was low ($I^2 = 32.6\%$, $p = 0.217$), and Egger's test demonstrated significant publication bias (Supplementary Information SI Table S3). Sensitivity testing found that one study (n = 1000) significantly influenced the results (Supplementary Information SI Figure S1), which demonstrated no difference in RBC content between the two groups (SMD = 0.002; 95% CI, −0.123–0.126). Omitting the study would not produce statistically significant results.

### 3.14. Influence of Bridging Thrombolysis on Fibrin Content

A total of nine studies [13,19,21,34,35,40,43,45,46] reported on the influence of bridging thrombolysis on fibrin content. Two studies [39,44] found an association with lower fibrin content, one study [49] with higher fibrin content, and six studies [13,19,21,34,43,46] with no significant association. There were four studies [13,19,21,34] with 1148 participants in the meta-analysis comparing fibrin content in patients who received bridging thrombolysis vs. direct EVT. The meta-analysis found that the fibrin fraction was lower in patients who received bridging thrombolysis (SMD = −0.109; 95% CI, −0.403–0.186, $p = 0.470$); however, this result was not statistically significant. Heterogeneity was low ($I^2 = 46.5\%$, $p = 0.133$), and Egger's test demonstrated significant publication bias (Supplementary Information SI Table S3). Sensitivity testing found that one study (n = 1000) significantly influenced the results (Supplementary Information SI Figure S1), which demonstrated no difference in fibrin content between the two groups (SMD = 0.036; 95% CI, −0.089–0.160). Omitting the study would not produce statistically significant results.

The summary effects of the meta-analyses are available in Supplementary Information SI Table S4. The forest plot for each meta-analysis is available in Figures 2–8. The graphs of the influence of a single study for each meta-analysis are available in Supplementary Information SI Figures S1–S7.

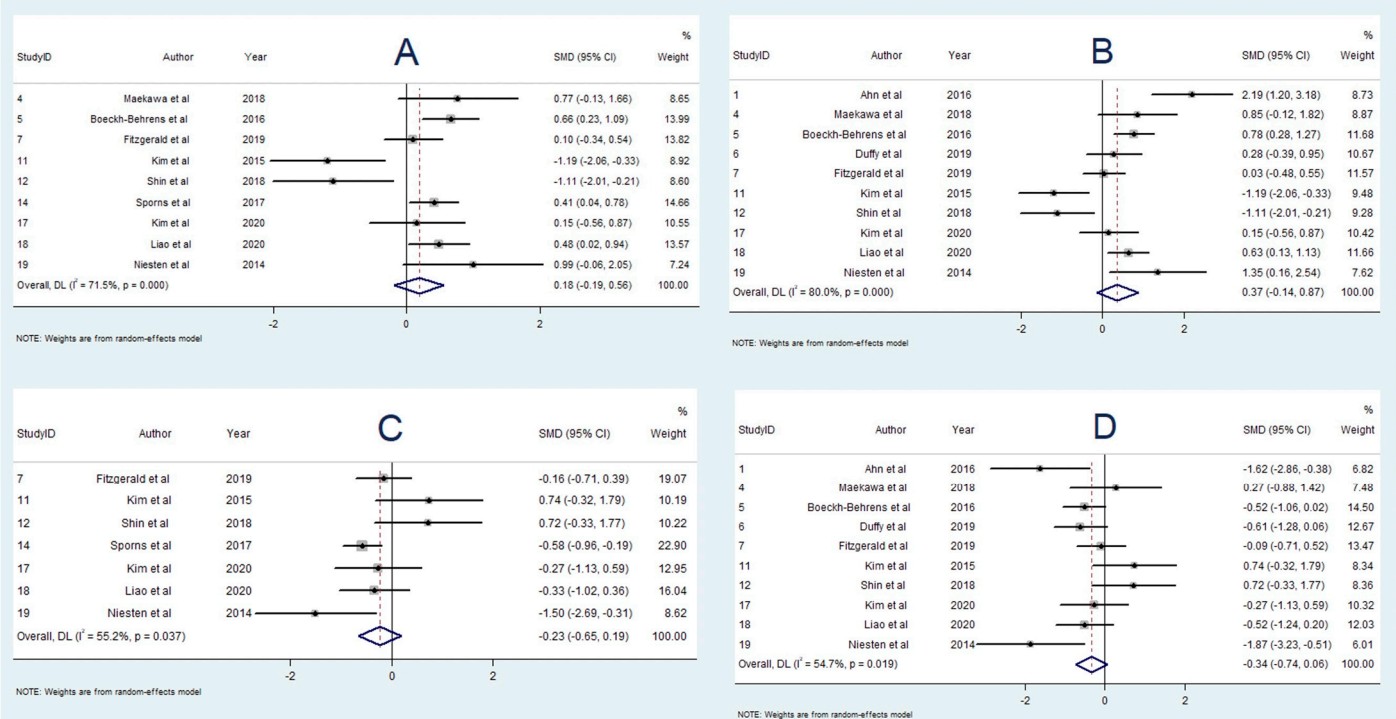

**Figure 2.** Forest Plots of Meta-analyses on RBC Content and Aetiology. (**A**): Non-cardioembolic vs. Cardioembolic Stroke. (**B**): LAA vs. Cardioembolic Stroke. (**C**): Cryptogenic vs. Non-cardioembolic stroke. (**D**): Cryptogenic vs. LAA stroke. Abbreviations: DL: DerSimonian and Laird method, SMD: Standarized Mean Difference, CI: Confidence Interval, RBC: Red Blood Cell, LAA: Large Artery Atherosclerosis.

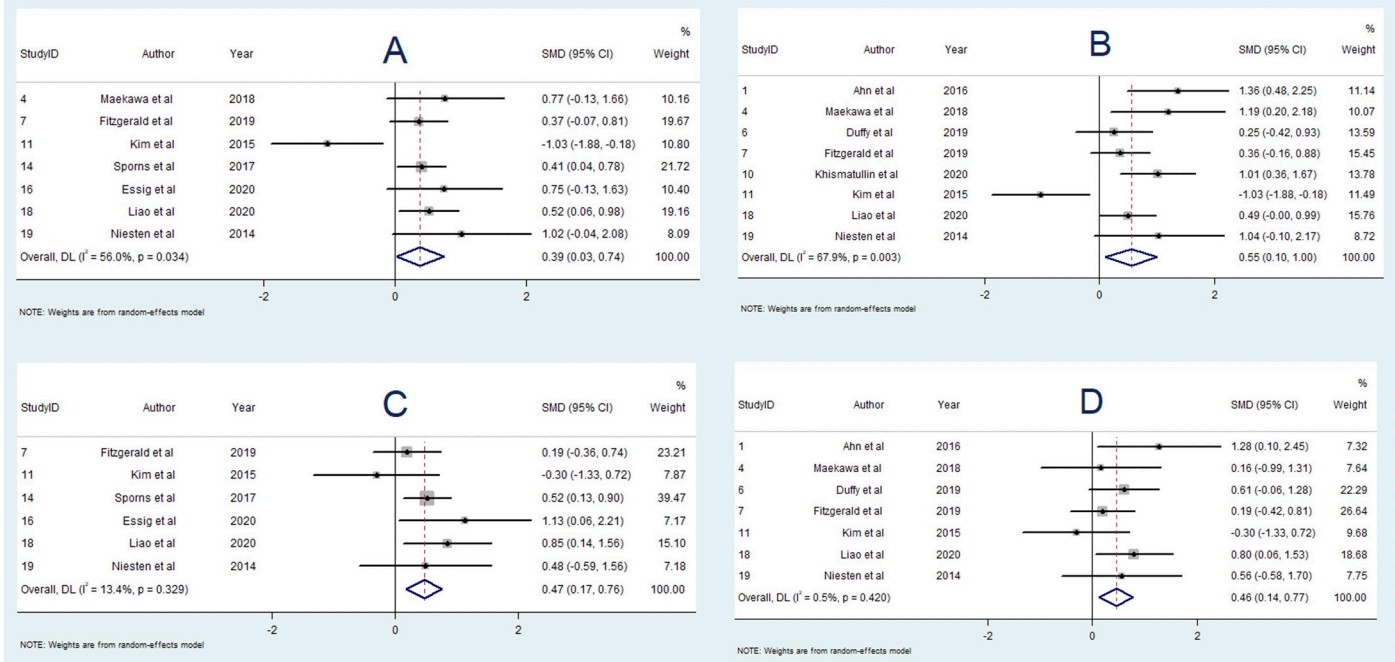

**Figure 3.** Forest Plots of Meta-analyses on Fibrin Content and Aetiology. (**A**): Cardioembolic vs. Non-cardioembolic stroke. (**B**): Cardioembolic vs. LAA stroke. (**C**): Cryptogenic vs. non-cardioembolic stroke. (**D**): Cryptogenic vs. LAA Stroke. Abbreviations: DL: DerSimonian and Laird method, SMD: Standarized Mean Difference, CI: Confidence Interval, LAA: Large Artery Atherosclerosis.

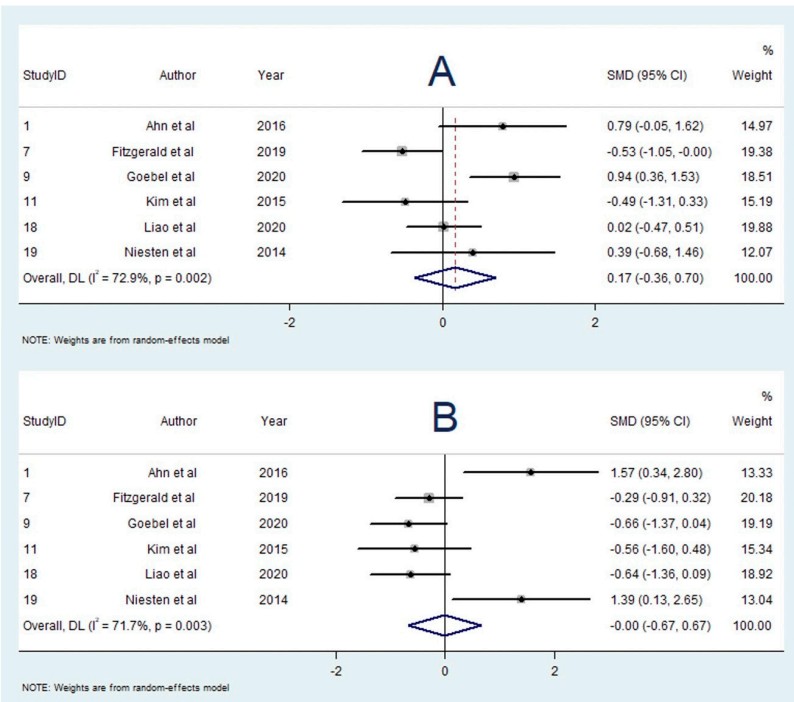

**Figure 4.** Forest Plots of Meta-analyses on Platelet Content and Aetiology. (**A**): Cardioembolic vs. LAA Stroke. (**B**): Cryptogenic vs. LAA Stroke. Abbreviations: DL: DerSimonian and Laird method, SMD: Standarized Mean Difference, CI: Confidence Interval, LAA: Large Artery Atherosclerosis.

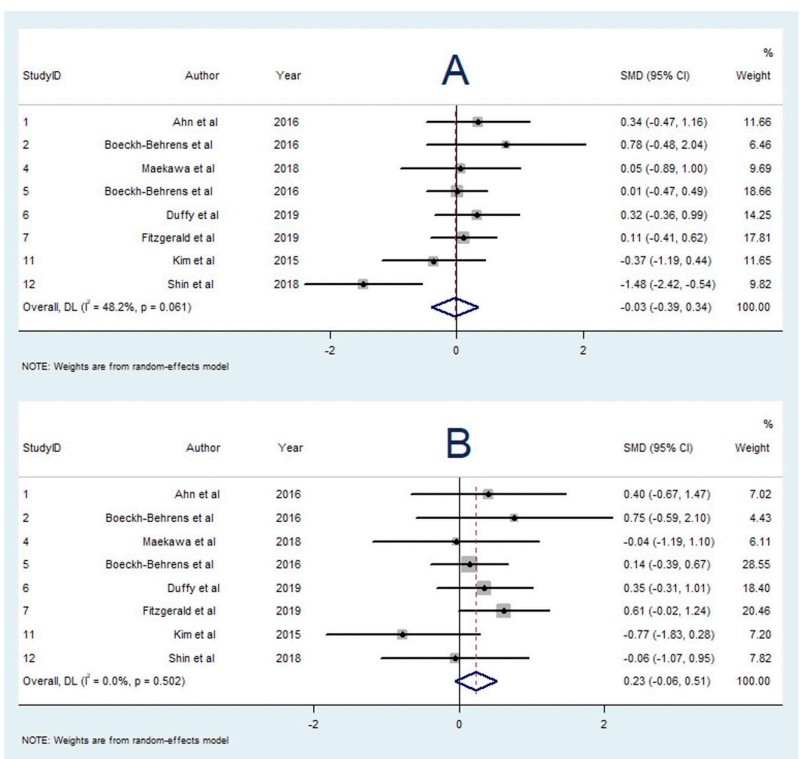

**Figure 5.** Forest Plots of Meta-analyses on WBC Content and Aetiology. (**A**): Cardioembolic vs. LAA Stroke. (**B**): Cryptogenic vs. LAA Stroke. Abbreviations: DL: DerSimonian and Laird method, SMD: Standarized Mean Difference, CI: Confidence Interval, WBC: white blood cell; LAA: Large Artery Atherosclerosis.

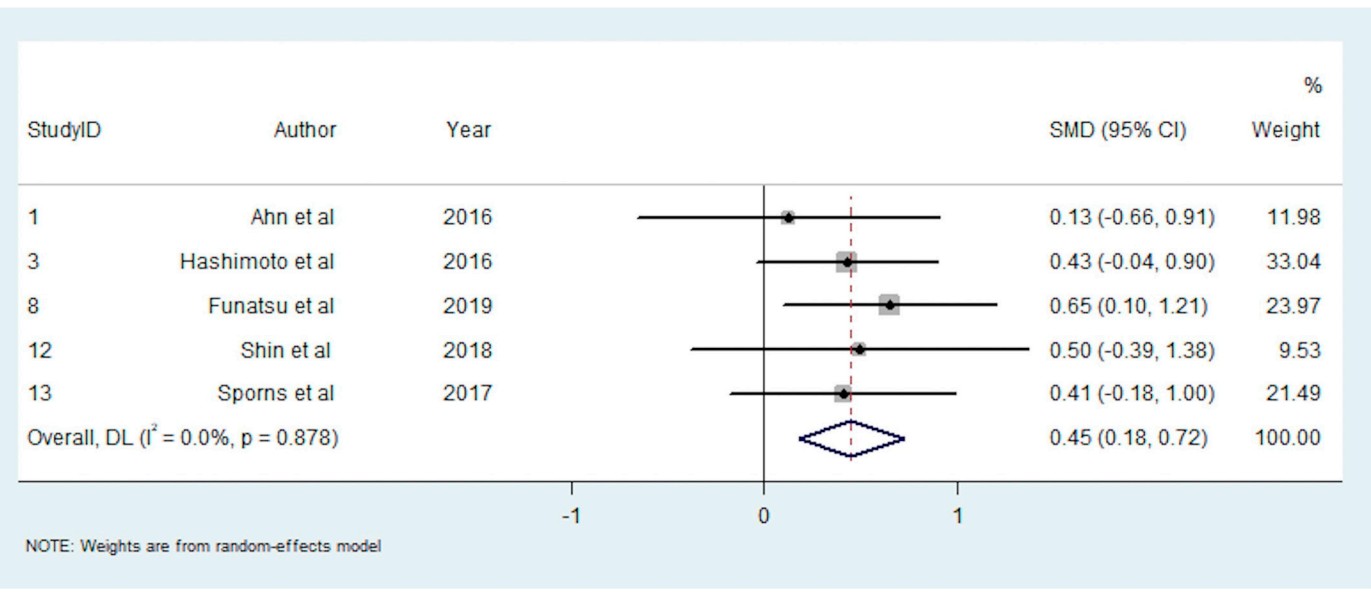

**Figure 6.** Forest Plot of Meta-analysis on RBC Content and Successful Recanalisation. Abbreviations: DL: DerSimonian and Laird method, SMD: Standarized Mean Difference, CI: Confidence Interval, RBC: Red Blood Cell.

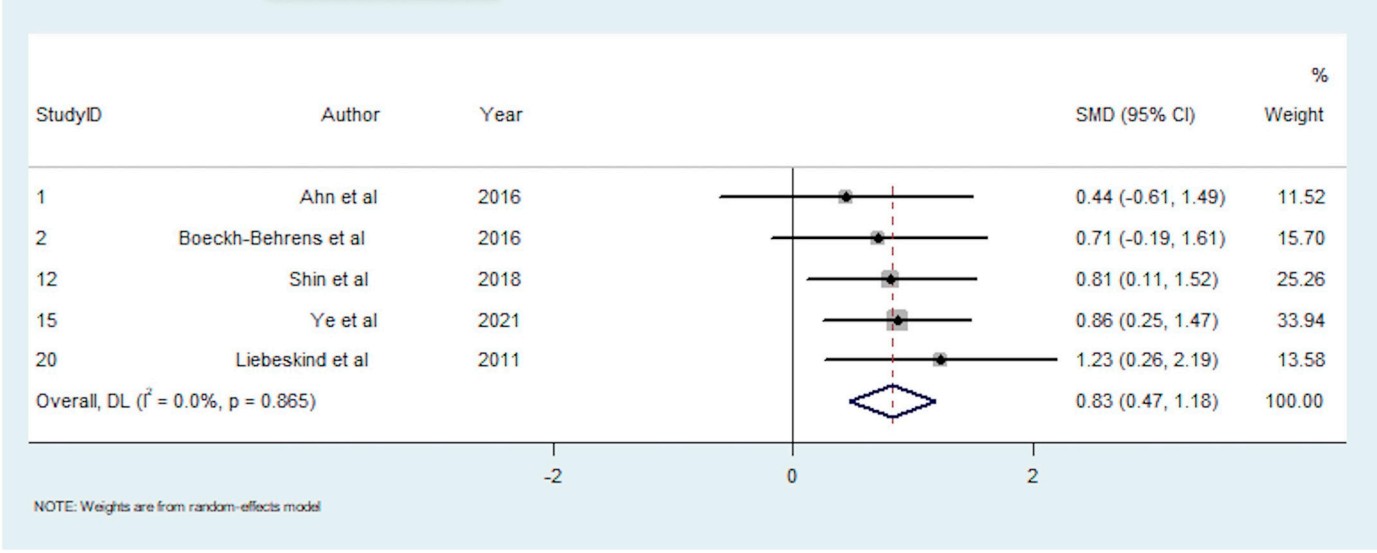

**Figure 7.** Forest Plot of Meta-analysis on RBC Content and Positive HMCAS. Abbreviations: DL: DerSimonian and Laird method, SMD: Standarized Mean Difference, CI: Confidence Interval, RBC: Red Blood Cell, HMCAS: Hyperdence Middle Cerebral Artery Sign.

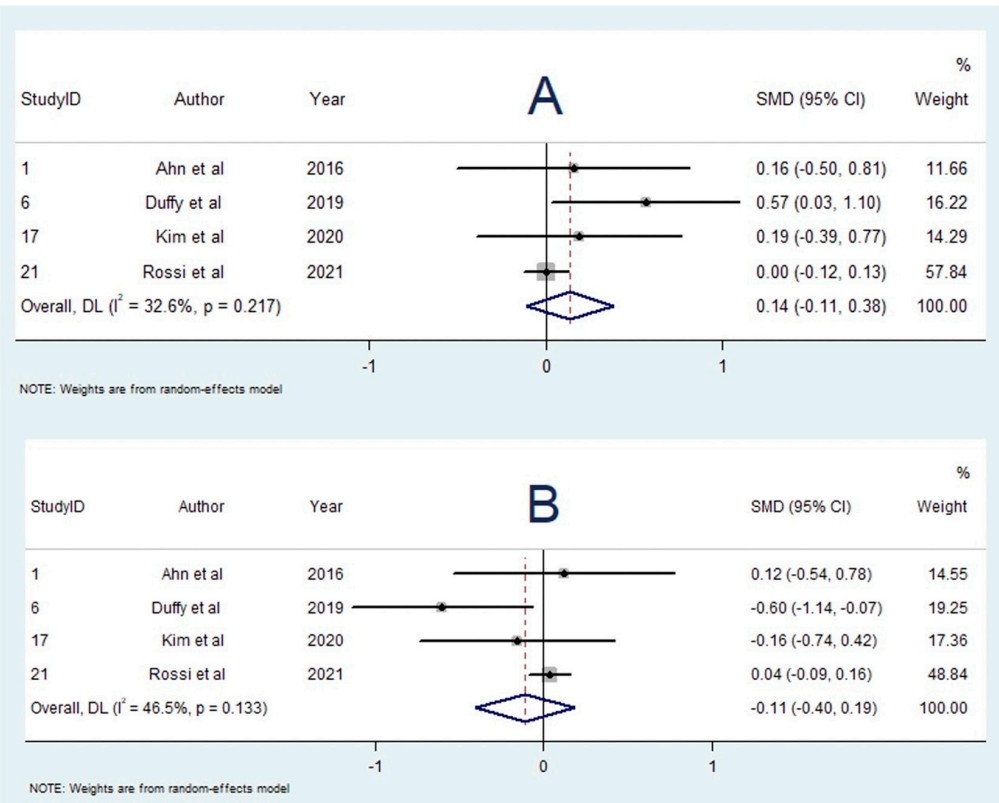

**Figure 8.** Forest Plots of Meta-analyses on clot composition and bridging thrombolysis. (**A**): RBC content and bridging thrombolysis. (**B**) Fibrin content and bridging thrombolysis. Abbreviations: DL: DerSimonian and Laird method, SMD: Standarized Mean Difference, CI: Confidence Interval, RBC: Red Blood Cell.

## 4. Discussion

This meta-analysis revealed several key findings regarding the relationship between brain thrombus composition and stroke aetiology, recanalisation and pre-interventional CT imaging. Both cardioembolic and cryptogenic strokes had a significantly higher proportion of fibrin than non-cardioembolic strokes and LAA strokes. The similarity in composition between cryptogenic and cardioembolic strokes suggests that many cryptogenic strokes may be reclassified as cardioembolic. There were no significant associations found between RBC content, platelet or WBC content and aetiology. In terms of successful recanalisation, RBC percentage was significantly greater in patients who achieved successful recanalisation (a TICI score of 2b–3). With regards to pre-interventional imaging signs, patients with a HMCAS present on CT were also found to have clots with greater RBC content. This study did not demonstrate an association between IVT (pre-EVT) and composition.

Delineation of stroke aetiology is crucial for optimal management of acute ischemic stroke patients [50]. The relationship between stroke aetiology and thrombus composition has been described by several studies with varying results. The association of thrombus composition with stroke aetiology is a subject of ongoing research and could aid management of stroke [51]. To our knowledge, this is presumably the first meta-analysis to demonstrate any significant association between brain clot composition and aetiology, with the finding that fibrin content is significantly greater in cardioembolic thrombi. This correlation is consistent with the majority of the literature, though a few studies have described the opposite, with Kim et al. in 2015 [25], and Shin et al. in 2018 [26] finding higher RBC fraction in cardioembolic strokes instead. Indeed, analysing the influence of a single study in these hypotheses showed that omitting these studies would yield a statistically significant association between RBC fraction and non-cardioembolic stroke aetiology. Nevertheless, these two studies [25,26] are supported by traditional ideas on

the pathophysiology behind the ways in which aetiology impacts clot composition, contrary to our findings. The high shear stress conditions of arterial blood flow which causes atherosclerosis, is thought to give rise to clots of greater fibrin and platelet proportion. On the other hand, cardioembolic thrombi, which most commonly occur in a low-flow environment such as stasis, are thought to trap more RBCs [25,49]. Thus, despite finding a significant association between fibrin content and cardioembolic strokes, the utility is limited in the lack of pathophysiological explanation and that no other clot components demonstrated statistical significance. Furthermore, the high shear stress in large arteries is thought to cause platelet aggregation on the vessel wall, which manifests as emboli with denser platelet content along the periphery [16,24]. Cardiac thrombi, on the other hand, are thought to produce platelet aggregates within fibrin-rich areas [16]. Following these observations, these qualitative structural features are vital in clot histological analysis on top of the quantitative analyses included in this meta-analysis.

While this study focused on RBC, fibrin, platelet and WBC content, the process of thrombus formation is much more complex than these four components. Recent studies have described in detail the role of von Willebrand factor, neutrophil extracellular traps and DNA in thrombi, especially in fibrin-rich areas of clots [52]. A peripheral fibrin 'shell' containing these extra components is thought to surround fibrin-rich areas, while RBC-rich areas are composed of a thinner fibrin meshwork [52]. However, another theory is that this 'shell' is a feature of all clots regardless of aetiology, while the inner core is highly variable [53]. Furthermore, thrombus formation and ageing have been suggested as a marker of aetiology. As a thrombus ages, it undergoes contraction and organisation, whereby contractile platelets act on fibrin and cause RBC-rich and fibrin-rich areas to separate [52]. This can cause changes in the composition of fibrin fibres as stroke thrombi develop [54], or the formation of polyhedrocyte RBCs [32]. Cardiogenic clots may form over a longer time, thus allowing more WBCs and other components to invade the clot [35]. This could explain why cardioembolic thrombi, being potentially older clots, seem to have greater platelet and WBC content, although this result was not confirmed in the present study.

Furthermore, this study has demonstrated a significant difference in fibrin composition between cryptogenic and non-cardioembolic or LAA strokes. Cryptogenic strokes, therefore, are likely to have a similar composition to cardioembolic strokes, and in many cases may be reclassified as cardioembolic [55]. Delineating strokes of cryptogenic origin is an important factor in the management of stroke patients, as appropriate treatment can be used to prevent secondary stroke [56], such as anticoagulation therapy for treating underlying atrial fibrillation or antiplatelets for LAA [20].

This meta-analysis confirmed the positive association between RBC content and improved recanalisation outcomes. Clots with higher RBC content have been found to be more responsive to IVT, presumably due to the thinner fibrin meshwork in these clots [57]. Furthermore, the additional components in fibrin-rich clots act as a barrier to successful reperfusion, rendering the clot stiffer, with greater adhesion to the vessel wall, and harder to aspirate or integrate [31,50,51]. On the other hand, RBC-rich clots have lower viscosity and friction [22,36], allowing better integration and reperfusion outcomes. However, the decreased stiffness associated with greater RBC content has also been found to correlate with clot migration [58].

This meta-analysis confirmed the positive association between RBC content and the HMCAS, a result which has been repeatedly demonstrated with little contradiction. A potential explanation of this relationship is that the increased haemoglobin content in RBC-rich clots increases attenuation [48]. Despite this finding, the clinical utility of this sign can be further investigated. For instance, determining which method of clot extraction harbours the best results for clots according to their composition could advance stroke management workflows, by informing clinicians about the optimal method to use. A recent meta-analysis by Bhambri et al. [59] found an association between direct aspiration techniques and RBC content, which could be a promising avenue.

**Table 1.** Baseline Characteristics of Included Studies.

| Study ID | Study | Design | No. of Centres | Cohort Size | Age, Mean (SD) | Male, n (%) | Histological Staining Method(s) | Thrombectomy Device(s) | TICI 2b–3, n (%) | HMCAS +, n (%) | IVT, n (%) | RBC, Mean % (SD) | TOAST, n | | | |
|---|---|---|---|---|---|---|---|---|---|---|---|---|---|---|---|---|
| | | | | | | | | | | | | | 1 | 2 | 4 | 5 |
| 1 | Ahn et al. (2016) [16] | Retrospective Cohort | 1 | 36 | 69.3 (8.6) | 24 (67) | H&E, MSB, CD42b | Penumbra System | 28 (78) | 31/35 a (89) | 20 (56) | 37 (17) | 8 | 22 | | 6 |
| 2 | Boeckh-Behrens et al. (2016a) [35] | Prospective Cohort | 1 | 34 | 79 (18–90) b | 13 (38) | H&E, EVG | Solitaire 4–20, Solitaire 6–30, Trevo, Trevo pro 4, or Penumbra 4 | 34 (100) | 18/29 a (62) | 16 (47) | 32 (23) | 3 | 16 | 6 | 9 |
| 3 | Hashimoto et al. (2016) [36] | Retrospective Cohort | 1 | 83 | 75.1 (9.6) | 52 (63) | H&E, Masson's Trichrome | Merci retriever, Penumbra system, Stent retrievers, ADAPT: Penumbra 5MAX ACE catheter | 58 (70) | | 50 (60) | 53 (24) | 8 | 64 | 1 | 10 |
| 4 | Maekawa et al. (2018) [17] | Retrospective Cohort | 1 | 43 | 76.6 (13.8) | 21 (49) | H&E | Solitaire stent, Trevo retriever | 42 (98) | | 20 (47) | 33 (27) | 5 | 30 | 1 | 7 |
| 5 | Boeckh-Behrens et al. (2016a) [20] | Retrospective Cohort | 1 | 137 c | 73 (18–92) b | 67 (49) | H&E | | | | 85 (62) | 43 (23) | 22 | 67 | 11 | 36 |
| 6 | Duffy et al. (2019) [22] | Retrospective Cohort | 1 | 60 | | | | Trevo (Stryker), Embotrap (Cerenovus), and Catch (Balt) | 54 (90) | | 38 (63) | 48 (20) | 15 | 20 | 3 | 22 |
| 7 | Fitzgerald et al. (2019b) [23] | Retrospective Cohort | >1 | 105 | 68 (25–93) b | | H&E, MSB | | 103 (98) | | 51 (49) | 41.9 | 20 | 52 | 12 | 21 |
| 8 | Funatsu et al. (2019) [37] | Retrospective Cohort | 1 | 101 | 74.9 (11.1) | 54 (53) | H&E, Mas son's Trichrome, EVG | ADAPT, Solitaire FR, XP ProVue Retriever, REVIVE SE, Solumbra catheter, Penumbra catheter | 86 (85) | | 41 (41) | | 11 | 79 | | 11 |
| 9 | Goebel et al. (2020) [39] | Retrospective Cohort | 1 | 85 | 72 (12.9) | 37 (44) | H&E, Ladewig trichrome, EVG, Von kossa, naphthol AS-D, chloroacetate, Prussian blue, CD68, CD45 | 5F Sofia distal access catheter, 6F Sofia Plus aspiration catheter, Penumbra catheter, Solitaire Stent retriever | 77 (91) | 43 (51) | 52 (61) | 41.7 | 16 | 51 | 1 | 17 |
| 10 | Khismatullin et al. (2020) [32] | Retrospective Cohort | 1 | 41 | 72 (1.5) | 24 (59) | H&E, scanning electron microscope | pRESET thrombectomy device, Catch retriever, Solitaire stent retriever, Penumbra aspiration system | | | 30 (73) | | 18 | 23 | | |

**Table 1.** *Cont.*

| Study ID | Study | Design | No. of Centres | Cohort Size | Age, Mean (SD) | Male, *n* (%) | Histological Staining Method(s) | Thrombectomy Device(s) | TICI 2b–3, *n* (%) | HMCAS +, *n* (%) | IVT, *n* (%) | RBC, Mean % (SD) | TOAST, *n* | | | |
|---|---|---|---|---|---|---|---|---|---|---|---|---|---|---|---|---|
| | | | | | | | | | | | | | 1 | 2 | 4 | 5 |
| 11 | Kim et al. (2015) [25] | Prospective Cohort | 1 | 37 | 69 (40–91) [b] | 20 (54) | H&E, CD61 | Solitaire Stent, Penumbra catheter | 31 (84) | | 23 (62) | 29 (29) | 8 | 22 | | 7 |
| 12 | Shin et al. (2018) [26] | Retrospective Cohort | 1 | 37 | 69.5 (14) | 20 (54) | H&E | Solitaire Stent retriever, Penumbra system | 31 (84) | 13/36 [a] (36) | 16 (43) | 32 (18) | 7 | 22 | | 8 |
| 13 | Sporns et al. (2017a) [38] | Cohort | 1 | 180 | 71 (15) | 92 (51) | H&E, EVG, Prussian Blue, CD3, CD20, CD68/KiM1P | pREset stent retriever | 168 (93) | | 120 (67) | 32 (29) | 34 | 74 | 11 | 60 |
| 14 | Sporns et al. (2017b) [21] | Retrospective Cohort | 1 | 187 | 71 (16) | 98 (52) | H&E, EVG, Prussian Blue, CD3, CD20, CD68/KiM1P | pREset stent retriever | 175 (94) | | 123 (66) | 32 (29) | 35 | 77 | 11 | 64 |
| 15 | Ye et al. (2021) [40] | Retrospective Cohort | 1 | 53 | 76 (14) | 26 (49) | H&E, MSB, VWF | Solumbra | 49 (92) | 37 (70) | 15 (28) | 33 (22) | 12 | 34 | | 7 |
| 16 | Essig et al. (2020) [34] | Retrospective Cohort | 1 | 37 | 65 (16) | 18 (49) | H&E, CD66b, Neutrophil elastase, H3Cit | | | | 26 (70) | | 7 [d] | 21 | | 9 |
| 17 | Kim et al. (2020) [24] | Retrospective Cohort | 1 | 52 | 62 (44) | 20 (38) | MSB, CD61, CD31, CD34 | Solitaire SR, Trevo SR, Penumbra catheter | 42 (81) | | 35 (67) | 17 (23) | 10 | 31 | | 11 |
| 18 | Liao et al. (2020) [19] | Retrospective Cohort | 1 | 88 | 63 (16) | 59 (67) | H&E, CD31 | | | | 23 (26) | 43 (14) | 25 | 46 | 6 | 11 |
| 19 | Niesten et al. (2014) [18] | Retrospective Cohort | 2 | 22 | 60 (13) | 11 (50) | H&E, Mallory's phosphotungstic acid-hematoxylin | Merci retriever, Trevo retriever, Solitaire stent | | | 17 (77) | 38 (19) | 8 | 6 | 3 | 5 |
| 20 | Liebeskind et al. (2011) [33] | Retrospective Cohort | 1 | 50 | 66 (21) | 26 (52) | H&E | Merci Retriever | | 10/20 (50) [a] | 7 (14) | 34 (21) | 66 | 21 | 26 | 33 |
| 21 | Rossi et al. (2021) [60] | Prospective Cohort | 4 | 1000 | | | MSB | | 893 (89) | | 451 (45) | 44 (25) | 221 | 346 | 55 | 255 [e] |

Abbreviations: ADAPT: A Direct Aspiration First Pass Technique, EVG: Elastica van Gieson, H&E: Haematoxylin and Eosin, HMCAS: Hyperdense Middle Cerebral Artery Sign, IVT: Intravenous Thrombolysis, MSB: Martius Scarlet Blue, RBC: Red Blood Cell, SD: standard deviation, TICI: Thrombolysis In Cerebral Infarction, TOAST: Trial of Org 10172 in Acute Stroke Treatment, vWF: von Willebrand Factor. [a] Data were not available for all patients. [b] Age reported as median (range) where mean (SD) was not available. [c] Contains 34 thrombi from Study 2. [d] 7 non-cardioembolic thrombi reported. [e] Aetiology was not reported in 123 patients.

**Table 2.** Clot Component Fractions according to Aetiology in Included Studies.

| Study ID | TOAST Study | RBC, Mean % (SD) | | | | Fibrin, Mean % (SD) | | | | Platelet, Mean % (SD) | | | | Fibrin/Platelet, Mean % (SD) | | | | WBC, Mean % (SD) | | | |
|---|---|---|---|---|---|---|---|---|---|---|---|---|---|---|---|---|---|---|---|---|---|
| | | 1 | 2 | 1 + 4 | 5 | 1 | 2 | 1 + 4 | 5 | 1 | 2 | 1 + 4 | 5 | 1 | 2 | 1 + 4 | 5 | 1 | 2 | 1 + 4 | 5 |
| 1 | Ahn et al. (2016) [16] | 60 (12) | 30 (12) | | 30 (22) | 23 (7) | 40 (14) | | 36 (14) | 17 (5) | 26 (13) | | 29 (11) | | | | | 4 (3) | 5 (3) | | 5 (3) |
| 2 | Boeckh-Behrens et al. (2016a) [35] | | | | | | | | | | | | | | | | | 5 (1) | 10 (7) | | 6 (2) |
| 4 | Maekawa et al. (2018) [17] | 51 (21) | 30 (26) | 50 (26) | 58 (33) | 33 (39) | 66 (26) | 46 (26) | 39 (32) | | | | | | | | | 4 (3) | 4 (3) | | 3 (5) |
| 5 | Boeckh-Behrens et al. (2016a) [20] | 56 (30) | 38 (20) | 53 (25) | 42 (21) | | | | | | | | | 36 (26) | 53 (19) | 41 (23) | 51 (21) | 6 (5) | 7 (4) | 9 (6) | 7 (5) |
| 6 | Duffy et al. (2019) [22] | 55 (19) | 49 (23) | | 44 (17) | 41 (16) | 46 (22) | | 51 (17) | | | | | | | | | 4 (3) | 5 (3) | | 5 (3) |
| 7 | Fitzgerald et al. (2019b) [23] | 42 (23) | 41 (24) | 44 (23) | 40 (20) | 33 (22) | 42 (25) | 33 (19) | 37 (21) | 22 (19) | 14 (14) | | 17 (15) | | | | | 3 (2) | 3 (2) | 3 (3) | 5 (5) |
| 9 | Goebel et al. (2020) [39] | | | | | | | | | 11 (11) | 20 (9) | | 14 (9) | | | | | | | | |
| 10 | Khismatullin et al. (2020) [32] | | | | | 13 (7) | 23 (11) | | | | | | | | | | | | | | |
| 11 | Kim et al. (2015) [25] | 8 (12) | 38 (28) | | 27 (34) | 52 (22) | 32 (18) | | 44 (30) | 35 (18) | 27 (16) | | 27 (8) | | | | | 5 (4) | 3 (4) | | 2 (2) |
| 12 | Shin et al. (2018) [26] | 18 (15) | 37 (17) | | 30 (17) | | | | | | | | | 76 (14) | 65 (17) | | 65 (17) | 6 (4) | 3 (1) | | 5 (3) |
| 14 | Sporns et al. (2017b) [21] | | 31 (32) | 45 (39) | 27 (25) | | 60 (30) | 47 (38) | 62 (25) | | | | | | | | | 9 (6) | 6 (5) | | 10 (7) |
| 16 | Essig et al. (2020) [34] | | | | | | 46 (30) | 26 (12) | 47 (22) | | | | | | | | | | | | |
| 17 | Kim et al. (2020) [24] | 22 (25) | 18 (23) | | 16 (21) | | | | | | | | | | | | | | | | |
| 18 | Liao et al. (2020) [19] | 45 (13) | 36 (15) | 43 (13) | 38 (14) | 30 (18) | 38 (17) | 29 (17) | 43 (15) | 25 (16) | 26 (13) | | 16 (12) | | | | | | | | |
| 19 | Niesten et al. (2014) [18] | 52 (17) | 29 (16) | 46 (17) | 21 (16) | 18 (9) | 33 (20) | 19 (10) | 24 (14) | 31 (12) | 37 (23) | | 55 (25) | | | | | | | | |

Abbreviations: RBC: Red Blood Cell, SD: standard deviation, TOAST: Trial of Org 10172 in Acute Stroke Treatment, WBC: White Blood Cell. Note: Number are presented as mean (standard deviation (SD)) of percentage (%).

**Table 3.** Clot Component Fractions and TICI Score.

| Study ID | Study | RBC, Mean % (SD) | | Fibrin, Mean % (SD) | | Platelet, Mean % (SD) | | Fibrin/Platelet, Mean % (SD) | | WBC, Mean % (SD) | |
|---|---|---|---|---|---|---|---|---|---|---|---|
| | | TICI 0–2a | TICI 2b–c | TICI 0–2a | TICI 2b–c | TICI 0–2a | TICI 2b–c | TICI 0–2a | TICI 2b–c | TICI 0–2a | TICI 2b–c |
| 1 | Ahn et al. (2016) [16] | 34 (20) | 36 (17) | 41 (18) | 34 (14) | 20 (10) | 26 (12) | | | 5 (4) | 4 (2) |
| 3 | Hashimoto et al. (2016) [36] | 47 (24) | 57 (23) | | | | | 48 (24) | 42 (22) | | |
| 8 | Funatsu et al. (2019) [37] | 42 (25) | 58 (24) | | | | | | | | |
| 12 | Shin et al. (2018) [26] | 24 (29) | 33 (15) | | | | | 71 (27) | 63 (14) | 5 (3) | 3 (2) |
| 13 | Sporns et al. (2017a) [38] | 21 (27) | 33 (29) | 70 (34) | 51 (30) | | | | | 10 (7) | 8 (5) |

Abbreviations: RBC: Red Blood Cell, SD: standard deviation, TICI: Thrombolysis in Cerebral Infarction, WBC: White Blood Cell. Note: Number are presented as mean (standard deviation (SD)) of percentage (%).

**Table 4.** Clot Component Fractions and HMCAS.

| Study ID | Study | RBC, Mean % (SD) | | Fibrin, Mean % (SD) | | Platelet, Mean % (SD) | | Fibrin/Platelet, Mean % (SD) | | WBC, Mean % (SD) | |
|---|---|---|---|---|---|---|---|---|---|---|---|
| | | HMCAS+ | HMCAS− | HMCAS+ | HMCAS− | HMCAS+ | HMCAS- | HMCAS+ | HMCAS− | HMCAS+ | HMCAS− |
| 1 | Ahn et al. (2016) [16] | 37 (19) | 29 (12) | 35 (15) | 34 (7) | 24 (12) | 30 (8) | | | 5 (3) | 7 (4) |
| 2 | Boeckh-Behrens et al. (2016a) [35] | 31 (23) | 16 (18) | | | | | | | | |
| 12 | Shin et al. (2018) [26] | 40 (10) | 26 (20) | | | | | 56 (9) | 69 (18) | 4 (2) | 4 (3) |
| 15 | Ye et al. (2021) [40] | 40 (23) | 21 (19) | 34 (16) | 44 (18) | 21 (15) | 30 (26) | | | | |
| 20 | Liebeskind et al. (2011) [33] | 47 (18) | 22 (23) | | | | | | | | |

Abbreviations: HMCAS: Hyperdense Middle Cerebral Artery Sign, RBC: Red Blood Cell, SD: standard deviation, WBC: White Blood Cell. Note: Number are presented as mean (standard deviation (SD)) of percentage (%).

**Table 5.** Clot component fractions and Bridging Thrombolysis.

| Study ID | Study | RBC, Mean % (SD) | | Fibrin, Mean % (SD) | | Platelet, Mean % (SD) | | WBC, Mean % (SD) | |
|---|---|---|---|---|---|---|---|---|---|
| | | IVT+ | IVT- | IVT+ | IVT− | IVT+ | IVT− | IVT+ | IVT− |
| 1 | Ahn et al. (2016) [16] | 37 (18) | 34 (18) | 37 (15) | 35 (15) | 23 (13) | 26 (11) | 4 (3) | 5 (3) |
| 6 | Duffy et al. (2019) [22] | 52 (18) | 41 (21) | 43 (17) | 54 (20) | | | 5 (3) | 5 (4) |
| 17 | Kim et al. (2020) [24] | 19 (22) | 14 (24) | 26 (16) | 30 (38) | 54 (14) | 50 (14) | | |
| 21 | Rossi et al. (2021) [60] | 44 (3) | 44 (27) | 30 (14) | 29 (17) | 19 (15) | 18 (14) | | |

Abbreviations: IVT: Intravenous Thrombolysis, RBC: Red Blood Cell, SD: standard deviation, WBC: White Blood Cell. Note: Number are presented as mean (standard deviation (SD)) of percentage (%).

While this meta-analysis did not demonstrate a statistically significant association, several studies have linked bridging thrombolysis with increased RBC content. This relationship would be expected, as the administration of r-tPA lyses fibrin, thereby raising the RBC fraction. However, a large study by Rossi et al. [60] found a similar histological composition between bridging thrombolysis and direct EVT groups. They suggested that r-tPA proportionally reduces the proportion of both fibrin and RBCs, overall rendering the clot smaller without significantly changing its composition. Again, qualitative changes such as the 'thinning' of the fibrin outer layer or other structural changes are at play [53], but were unable to be analysed by this meta-analysis. If RBC is significantly reduced or other properties are ascertained, this could inform stroke management by informing which method of EVT to use in patients who have received IVT.

## 5. Limitations

This study has several limitations which merit consideration. Firstly, beyond histopathological characterisation of intracranial brain clots, single-cell sequencing and immunohistological methods have also been applied to study specific features of clots [61,62]. However, this study focused on the morphological features of brain clots retrieved after EVT. Secondly, several aspects of the stroke clinical workup were not standardised or controlled, such as IVT administration, EVT method and histological analysis method. Besides, there are variations in brain thrombus collection, storage, analysis and reporting across studies and centres which may influence the underlying observations. Selection bias also inherently exists, as only patients with retrievable clots can be examined. Those who underwent successful IVT, or whose thrombi were unable to be extracted during EVT, could not be included in any cohort. In terms of data extraction for this meta-analysis, many studies were unable to be included as their thrombus composition results were reported as being "RBC-rich", or "fibrin-rich", dichotomising composition rather than considering it as a continuous variable. Since each study uniquely defined these terms, multiple were excluded from this meta-analysis, limiting the data. Furthermore, qualitative observations could not be included in this quantitative analysis, although many pertinent findings exist in this aspect. Similarly, a few relevant hypotheses could not be assessed as the minimum number of studies for meta-analysis was not reached. In terms of the data, studies that reported results in median (IQR) form were converted to mean (SD) form. Although a verified method of conversion was used, this may introduce a slight error in the data used. Lastly, Egger's test demonstrated significant publication bias for many studies included in the meta-analyses. Despite this limitation, the use of random-effects modelling, used consistently to test various hypotheses, would have presumably mitigated some of this bias.

## 6. Conclusions

In conclusion, this meta-analysis found that fibrin composition is significantly higher in strokes of cardioembolic and cryptogenic origin and that RBC content is positively associated with the HMCAS and better reperfusion outcomes in AIS patients treated with EVT. Important advances to stroke clinical workup can be derived from these findings, in which several aspects remain to be optimised. As data are still limited in terms of several thrombus components as well as a standardised method of analysis, further studies are required to validate these findings and assess their clinical utility.

**Supplementary Materials:** The following supporting information can be downloaded at: https://www.mdpi.com/article/10.3390/neurolint14040063/s1, Figure S1: Influence of a single study in meta-analysis estimation: Clot Composition and Bridging Thrombolysis; Figure S2: Influence of a single study in meta-analysis estimation: RBC and Aetiology; Figure S3: Influence of a single study in meta-analysis estimation: Fibrin and Aetiology; Figure S4: Influence of a single study in meta-analysis estimation: Platelet and Aetiology; Figure S5: Influence of a single study in meta-analysis estimation: WBC and Aetiology; Figure S6: Influence of a single study in meta-analysis estimation: RBC and TICI Score; Figure S7: Influence of a single study in meta-analysis estimation: Clot Composition and

Bridging Thrombolysis; Table S1: Modified Jadad Analysis for Methodological Quality; Table S2: Baseline characteristics of studies excluded from meta-analysis; Table S3: Egger's test for publication bias assessment of the included studies; Table S4: Summary Effects of meta-analysis.

**Author Contributions:** S.M.M.B. conceived the study, contributed to the planning, draft, and revision of the manuscript; supervision of the students. S.M.M.B. encouraged J.H. to investigate and supervised the findings of this work. J.H. and S.M.M.B. wrote the first draft of this paper. M.C.K. contributed to the supervision of the student, validation, and revision of the manuscript. All authors have read and agreed to the published version of the manuscript.

**Funding:** This research received no funding.

**Institutional Review Board Statement:** Not applicable.

**Informed Consent Statement:** Not applicable.

**Data Availability Statement:** The original contributions presented in the study are included in the article/Supplementary Information, further inquiries can be directed to the corresponding author.

**Acknowledgments:** Funding grant for the NSW Brain Clot Bank (Chief Investigator: S.M.M.B.) from the NSW Ministry of Health (2019–2022) is acknowledged. The funding body has no role in the study design, data collection, analysis, interpretation of findings and manuscript preparation. The content is solely the responsibility of the authors and does not necessarily represent the official views of the affiliated/funding organisation.

**Conflicts of Interest:** The authors declare that they have no conflict of interest.

## Abbreviations

| | |
|---|---|
| AIS | Acute Ischaemic Stroke |
| IVT | Intravenous Thrombolysis |
| EVT | Endovascular Thrombectomy |
| RBC | Red Blood Cell |
| LAA | Large Artery Atherosclerosis |
| HMCAS | Hyperdense Middle Cerebral Artery Sign |
| PRISMA | Preferred Reporting Items for Systematic Reviews and Meta-Analyses |
| TICI | Thrombolysis in Cerebral Infarction |
| WBC | White Blood Cell |
| SMD | Standard Mean Difference |
| TOAST | Trial of Org 10,172 in Acute Stroke Treatment |
| SVS | Susceptibility Vessel Sign |
| r-tPA | Recombinant Tissue Plasminogen Activator |
| H&E | Haematoxylin and Eosin |
| MSB | Martius Scarlet Blue |
| EVG | Elastica van Gieson |

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
