# Peer review of "Is Composition of Brain Clot Retrieved by Mechanical Thrombectomy Associated with Stroke Aetiology and Clinical Outcomes in Acute Ischemic Stroke?—A Systematic Review and Meta-Analysis"

_2035-8377, doi:10.3390/neurolint14040063_

Round 1

Reviewer 1 Report

In the captions of tables 1-5 there are abbreviations that are explained, their order should be alphabetical

In the tables, the abbreviation for the statistical analysis "SD" is used - it should also be explained.

The caption under Tables 1-5 should indicate the type of statistical analysis / test that was used for the analysis

The abstract should be rewritten - the use of a large number of abbreviations in the abstract should be avoided

Author Response

We thank the reviewer for the review of our work and the comments provided. We provide point by point rebuttal to comments provided.

C#1: In the captions of tables 1-5 there are abbreviations that are explained, their order should be alphabetical

Reply# All the abbreviations have now been alphabetically arranged. Thank you for pointing this.

C#2: In the tables, the abbreviation for the statistical analysis "SD" is used - it should also be explained.

Reply# We have now explained the SD

C#3: The caption under Tables 1-5 should indicate the type of statistical analysis / test that was used for the analysis

Reply# Statement to that effect has been added.

C#4: The abstract should be rewritten - the use of a large number of abbreviations in the abstract should be avoided

Reply# The abstract has been revised and abbreviations have been expanded.

Reviewer 2 Report

In the present meta-analysis including 21 relevant papers suggested an interesting finding: fibrin composition seems to be higher in cardioembolic/cryptogenic strokes, whereas RBC content is positively associated with the hyperdense middle cerebral artery sign (HMCAS) and better reperfusion outcomes. Although the clinical application of these finding is challenging, overall, the paper is well-written and -designed. All figures and tables are informative. The discussion is also relevant to the findings. The authors also nicely discussed about the limitations of the study in section 5. I believe the paper in the current format merit publication.

Author Response

We thank the reviewer for review of our work and for providing constructive feedback. Replies to questions below;

C#1: In the present meta-analysis including 21 relevant papers suggested an interesting finding: fibrin composition seems to be higher in cardioembolic/cryptogenic strokes, whereas RBC content is positively associated with the hyperdense middle cerebral artery sign (HMCAS) and better reperfusion outcomes. Although the clinical application of these finding is challenging, overall, the paper is well-written and -designed. All figures and tables are informative. The discussion is also relevant to the findings. The authors also nicely discussed about the limitations of the study in section 5. I believe the paper in the current format merit publication

Reply# Thank you for the positive review of our work and appreciate the observations made in the report.

Reviewer 3 Report

This study aimed to perform a systematic review and meta-analysis to delineate the association of brain clot composition with stroke aetiology and post-reperfusion outcomes in patients receiving endovascular thrombectomy. The authors conducted a systematic literature review and meta-analysis by extracting data from several research databases (MEDLINE/PubMed, Cochrane, and Google Scholar) published since 2010. They used appropriate key search terms to identify clinical studies concerning stroke thrombus composition, aetiology, and clinical outcomes, in accordance with Preferred Reporting Items for Systematic Reviews and Meta-Analyses (PRISMA) guidelines. The authors identified 30 articles reporting on the relationship between stroke thrombus composition or morphology and aetiology, imaging, or clinical outcomes, of which 21 were included in the meta-analysis. The study found that strokes of cardioembolic origin (SMD = 0.388; 95% CI, 0.032 – 0.745) and cryptogenic origin (SMD = 0.468; 95% CI, 0.172 – 0.765) had significantly higher fibrin content than strokes of non-cardioembolic origin. Large artery atherosclerosis (LAA) strokes had significantly lower fibrin content than cardioembolic (SMD = 0.552; 95% CI, 0.099 – 1.004) or cryptogenic (SMD = 0.455; 95% CI, 0.137 – 0.774) strokes. Greater red blood cell (RBC) content was also significantly associated with a thrombolysis in cerebral infarction (TICI) score of 2b-3 (SMD= 0.450; 95% CI, 0.177 – 0.722), and a positive hyperdense middle cerebral artery sign (HMCAS) (SMD = 0.827; 95% CI, 0.472 – 1.183). No significant associations were found between RBC, platelet, or white blood cell (WBC) content and aetiology, or between clot composition and bridging thrombolysis. 

The article is well written. Nevertheless, the introduction section lacks current references emphasizing the importance of factors other than just the embolic factor itself, e.g. Benjamin et al and Wańkowicz et al.

Author Response

We thank the reviewer for review of our work and constructive feedback.

We have made changes to the manuscript as per the comments provided.

Please see point-by-point rebuttal below;

1. This study aimed to perform a systematic review and meta-analysis to delineate the association of brain clot composition with stroke aetiology and post-reperfusion outcomes in patients receiving endovascular thrombectomy. The authors conducted a systematic literature review and meta-analysis by extracting data from several research databases (MEDLINE/PubMed, Cochrane, and Google Scholar) published since 2010. They used appropriate key search terms to identify clinical studies concerning stroke thrombus composition, aetiology, and clinical outcomes, in accordance with Preferred Reporting Items for Systematic Reviews and Meta-Analyses (PRISMA) guidelines. The authors identified 30 articles reporting on the relationship between stroke thrombus composition or morphology and aetiology, imaging, or clinical outcomes, of which 21 were included in the meta-analysis. The study found that strokes of cardioembolic origin (SMD = 0.388; 95% CI, 0.032 – 0.745) and cryptogenic origin (SMD = 0.468; 95% CI, 0.172 – 0.765) had significantly higher fibrin content than strokes of non-cardioembolic origin. Large artery atherosclerosis (LAA) strokes had significantly lower fibrin content than cardioembolic (SMD = 0.552; 95% CI, 0.099 – 1.004) or cryptogenic (SMD = 0.455; 95% CI, 0.137 – 0.774) strokes. Greater red blood cell (RBC) content was also significantly associated with a thrombolysis in cerebral infarction (TICI) score of 2b-3 (SMD= 0.450; 95% CI, 0.177 – 0.722), and a positive hyperdense middle cerebral artery sign (HMCAS) (SMD = 0.827; 95% CI, 0.472 – 1.183). No significant associations were found between RBC, platelet, or white blood cell (WBC) content and aetiology, or between clot composition and bridging thrombolysis. 

The article is well written. Nevertheless, the introduction section lacks current references emphasizing the importance of factors other than just the embolic factor itself, e.g. Benjamin et al and Wańkowicz et al.

Reply# We thank the reviewer for the positive review of our work and comments. We have added the said references and also added the following statement to the introduction. 

Beyond embolism, other factors such as atrial fibrillation and HIV infection may also cause stroke [1,2].

References:

  1. Benjamin, L.A.; Bryer, A.; Emsley, H.C.A.; Khoo, S.; Solomon, T.; Connor, M.D. HIV infection and stroke: current perspectives and future directions. The Lancet Neurology 2012, 11, 878-890, doi:https://doi.org/10.1016/S1474-4422(12)70205-3.
  2. Wańkowicz, P.; Nowacki, P.; Gołąb-Janowska, M. Atrial fibrillation risk factors in patients with ischemic stroke. Arch Med Sci 2021, 17, 19-24, doi:10.5114/aoms.2019.84212.

Round 2

Reviewer 3 Report

I accept the article in the current version.

Author Response

Thank you for favourable review of our work